# Evaluation of Two Recombinant Protein-Based Vaccine Regimens against *Campylobacter jejuni*: Impact on Protection, Humoral Immune Responses and Gut Microbiota in Broilers

**DOI:** 10.3390/ani13243779

**Published:** 2023-12-07

**Authors:** Noémie Gloanec, Muriel Guyard-Nicodème, Raphaël Brunetti, Ségolène Quesne, Alassane Keita, Marianne Chemaly, Daniel Dory

**Affiliations:** 1GVB—Viral Genetics and Biosafety Unit, French Agency for Food, Environmental and Occupational Health & Safety (ANSES), 22440 Ploufragan, France; noemie.gloanec@gmail.com (N.G.); raphael.brunetti95@gmail.com (R.B.); daniel.dory@anses.fr (D.D.); 2HQPAP—Unit of Hygiene and Quality of Poultry and Pork Products, French Agency for Food, Environmental and Occupational Health & Safety (ANSES), 22440 Ploufragan, France; segolene.quesne@anses.fr (S.Q.); marianne.chemaly@anses.fr (M.C.); 3Life Environmental Sciences Department, University of Rennes 1, 37500 Rennes, France; 4SELEAC—Avian Breeding and Experimental Department, French Agency for Food, Environmental and Occupational Health & Safety (ANSES), 22440 Ploufragan, France; alassane.keita@anses.fr

**Keywords:** *Campylobacter jejuni* caecal colonisation, humoral immune response, caecal microbiota composition

## Abstract

**Simple Summary:**

*Campylobacter* is the most common cause of human bacterial gastroenteritis, and poultry products are the main source of exposure. To reduce human campylobacteriosis, it is necessary to reduce the contamination level of *Campylobacter* in live poultry. Vaccination could be a solution, but no vaccines for *Campylobacter* are available to date. In our previous study, a vaccine candidate induced partial protection against *Campylobacter* in broilers. The objective of the present study was to evaluate whether a protein-based vaccine inoculated at two different frequencies (two or four times) would help to improve the protective immune response during a 42-day trial in broilers orally infected by the bacterium. A specific antibody response was observed regardless of the frequency of inoculation. Moreover, microbiota analysis revealed that significant differences were observed between the groups, but that vaccination did not alter the relative abundance of the main bacterial taxa residing in the caeca. No reduction in *Campylobacter* caecal load was observed regardless of the vaccine regimen tested. Additional studies testing other vaccine candidates are needed to develop an effective vaccine against *Campylobacter* in broilers.

**Abstract:**

*Campylobacter* infections in humans are traced mainly to poultry products. While vaccinating poultry against *Campylobacter* could reduce the incidence of human infections, no vaccine is yet available on the market. In our previous study using a plasmid DNA prime/recombinant protein boost vaccine regimen, vaccine candidate YP437 induced partial protective immune responses against *Campylobacter* in broilers. In order to optimise vaccine efficacy, the vaccination protocol was modified using a protein prime/protein boost regimen with a different number of boosters. Broilers were given two or four intramuscular protein vaccinations (with the YP437 vaccine antigen) before an oral challenge by *C. jejuni* during a 42-day trial. The caecal *Campylobacter* load, specific systemic and mucosal antibody levels and caecal microbiota in the vaccinated groups were compared with their respective placebo groups and a challenge group (*Campylobacter* infection only). Specific humoral immune responses were induced, but no reduction in *Campylobacter* caecal load was observed in any of the groups (*p* > 0.05). Microbiota beta diversity analysis revealed that the bacterial composition of the groups was significantly different (*p* ≤ 0.001), but that vaccination did not alter the relative abundance of the main bacterial taxa residing in the caeca. The candidate vaccine was ineffective in inducing a humoral immune response and therefore did not provide protection against *Campylobacter* spp. infection in broilers. More studies are required to find new candidates.

## 1. Introduction

*Campylobacter* spp. are microaerophilic Gram-negative, spiral-shaped bacteria and the principal cause of bacterial foodborne zoonoses in the European Union, with 127,840 human cases in 2021 [1]. *C. jejuni* is the main species involved, causing approximately 88% of these campylobacteriosis cases. The main reservoir of *Campylobacter* is poultry, with an estimated prevalence of 71.2% and 75.8% in broiler batches and in broiler carcasses, respectively, in France [2]. Consequently, the majority of human infections—usually characterised by a gastrointestinal illness—can be attributed to the consumption of *Campylobacter*-contaminated broiler meat products [1] and cross-contamination during the handling of meat in cooking environments [3,4].

Potential strategies to control *Campylobacter* should be implemented at the farm level because of its impact throughout the broiler food chain (slaughter, retail sales and consumption). *Campylobacter* primarily colonises the caeca, where the *Campylobacter* load generally reaches up to about 8 log_10_ CFU/g of caecal contents [5,6]. Reducing *Campylobacter* in the chicken reservoir, and more specifically in the caeca of chickens, would be an effective way to reduce the risk to consumers and the public health burden, which has been estimated at about 2.4 billion euros each year in the European Union, based on 2011 figures [7]. In fact, it has been estimated that either a 2 log_10_ or 3 log_10_ reduction in broiler caecal concentrations would reduce the relative European Union risk of human campylobacteriosis attributable to broiler meat by either 42% or 58%, respectively [8]. Several control strategies have been tested at the broiler primary production level, such as hygiene and biosecurity measures or nutritional and immune strategies [9]. However, despite promising results, there is currently no efficient strategy for reducing *Campylobacter*’s colonisation of the avian gut. One of the promising strategies is the vaccination of broilers, and several vaccination strategies have been investigated in recent decades [10,11,12,13,14,15,16,17,18,19,20,21,22,23]. While initial studies used whole-cell vaccines, more recent studies have tested subunit vaccines. Moreover, different routes of immunisation have been tested, such as intramuscular, subcutaneous, oral and in ovo. However, the results of such vaccination studies have been inconsistent, and there is currently no vaccine on the market. Although *C. jejuni* is considered to be a commensal in the avian gut [24], several studies have shown that *C. jejuni* can induce immune responses (e.g., production of pro- and anti-inflammatory cytokines and β-defensins, and stimulation of immune cells) and changes in the composition of the microbiota [25,26,27,28,29,30]. Furthermore, recent work has focused on the impact of *Campylobacter* vaccination in chickens on immune responses and the gut microbiota [11,31,32].

In our previous study [33], protein WP_002869420.1 (a haemolysin-secreting/-activating protein of the ShlB/FhaC/HecB family), named YP437 here, was identified by reverse vaccinology as a potential vaccine candidate against *Campylobacter* spp. in broilers. In the first in vivo study, a plasmid DNA prime/recombinant protein boost vaccine regimen [34] induced a mean reduction of about 3.6 log_10_ of caecal *Campylobacter* load accompanied by a production of systemic antibodies against *Campylobacter* in broilers. However, this reduction in *Campylobacter* load was not observed in two further trials [34,35].

In order to find a strategy to improve the protective effect of this vaccine in broilers, the present study focused on the impact of inoculation frequency using the protein form of vaccines.

Therefore, the objectives of this project in broilers were (1) to evaluate the efficacy of the vaccine candidate using a protein prime/protein boost regimen; (2) to assess the impact of the frequency (booster inoculations) of vaccine inoculations; and (3) to study the systemic and local humoral immune responses generated and gut microbiota modifications after vaccination in greater depth.

## 2. Materials and Methods

### 2.1. Production of the Recombinant YP437 Protein Vaccines

The recombinant YP437 protein was produced as described previously [34]. The emulsifications for the intramuscular route were performed 2–5 days before vaccinations. For each chicken, 100 µg of recombinant YP437 protein or phosphate-buffered saline (PBS) for the vaccinated groups and placebo groups, respectively, was emulsified in adjuvant MONTANIDE^TM^ ISA 78 VG (37/63, *w*/*w*) using the Ultra Turrax^®^ Tube Drive Basic emulsification system (IKA^®^-Werke GmbH, Staufen, Germany) according to the manufacturer’s recommendations, and then stored at 4 °C.

### 2.2. Campylobacter Strain and Growth Conditions

The *C. jejuni* C97ANSES640 strain was used for the oral challenge as described previously [31].

### 2.3. Avian Vaccine Experiment

The trial was carried out at the Animal Biosafety Level 2 facilities of ANSES’s Ploufragan Laboratory (France), an approved facility for animal experimentation (No. E-22-745-1). The procedure was validated by the French Ministry of Higher Education, Research and Innovation (APAFIS#33555-2021102215327824-v3). A total of 97 day-of-hatch conventional Ross 308 broiler chicks (males and females), vaccinated against infectious bronchitis, were purchased from a local hatchery. At the beginning of the experiment, the absence of *Campylobacter* spp. was confirmed in husbandries (including the feeding and drinking systems) and in five chicks according to the standard NF EN ISO 10272-1 (2017) [36]. The remaining chicks were randomly divided (from 19 to 20 chicks, Figure 1) into five groups (YP437 I2, P I2, YP437 I4, P I4, challenge, Figure 1) and were reared in 3.42 m^2^ floor pens (1.85 × 1.85 m^2^) as described in Figure 1.

Intramuscular vaccinations (0.3 mL) in the thigh were performed on day 5 (D5) and D12 for the YP437 I2, P I2, YP437 I4 and P I4 groups and additionally on D19 and D28 for the YP437 I4 and P I4 groups using 26 G needles. All the chickens were challenged orally with 10^4^ CFU of *C. jejuni* on D19 [31,37]. On D19, 5 birds per group were euthanised (electronarcosis followed by bleeding) and on D42, 14 or 15 birds per group were similarly euthanised. Blood (collected during bleeding), bile and caeca (collected during necropsy) were processed as previously described [31] through to determination of the humoral immune response, *Campylobacter* spp. caecal enumeration and caecal microbiota analysis. Each bird was individually weighed on days 5, 12, 19, 28 and 42. The experimental design is summarised in Figure 1.

The facilities (closed building without windows) were equipped with programmable electric lights, automated electric heating and forced ventilation. Lighting was 23 h per day from day 1 to day 7 and then 18 h per day for the rest of the experiment. The environmental temperature was gradually reduced from 32 °C on D1 to 18 °C on D35. Litter was composed of unused wood shavings. The birds were fed standard diets that met or exceeded nutritional requirements for broilers, formulated and manufactured by a commercial feed mill. They received water and feed ad libitum. A starter–grower diet was distributed from D1 to D22, then a grower–finisher diet was used until D42. Daily observations of the birds were conducted to ensure that no adverse reactions occurred.

### 2.4. Campylobacter Caecal and Faecal Enumeration

*Campylobacter* enumerations for caecal contents were performed after direct plating on mCCDA after incubation for 48 h at 41.5 °C under microaerobic conditions according to the decimal dilution method in tryptone salt broth as described previously [31]. In the same way, three days post-inoculation on D22, *Campylobacter* enumerations in chickens were checked from pooled faeces in each group.

### 2.5. Specific Systemic and Mucosal Immune Responses by Specific ELISAs

The levels of specific humoral response in serum and bile were measured using ELISA according to the previous study [31], except for the following modifications. The plates were coated with 0.5 μg/mL of purified YP437 recombinant protein and incubated with 1:100 of tested sera and bile followed by 1:25,000 diluted goat anti-chicken IgY– horseradish peroxidase (HRP) antibodies (Abcam, Paris, France) or by 1:5000 diluted goat anti-chicken IgA–HRP (Abcam, Paris, France). Each sample was measured in duplicate.

### 2.6. Statistical Analyses

R software (version 4.0.3) was used for statistical analyses [38]. To compare body weights, *Campylobacter* loads and specific antibody levels between groups, the ANOVA parametric test was used as the data were normally distributed and the variance was homogenous (checked by the Shapiro–Wilk normality test and Bartlett’s test, respectively) followed by the Tukey test; otherwise, the non-parametric Kruskal–Wallis test was performed, followed by the Wilcoxon test. Differences between groups were significant when the *p* values were lower than or equal to 0.05 (*p* ≤ 0.05).

### 2.7. DNA Extraction and PCR Amplification of 16S rRNA Gene Sequences and Microbiota Diversity Analysis

#### 2.7.1. DNA Extraction

Bacterial DNA was isolated from caecal pellets using the NucleoMag Tissue Kit (Macherey-Nagel, Hoerdt, France) according to the manufacturer’s protocol as described previously [31] and stored at −70 °C.

#### 2.7.2. Sequencing of the V3/V4 Variable Region of the 16S Ribosomal Genes

The V3–V4 variable region of the 16S rDNA gene was amplified by PCR (forward primer: 5′-TCGTCGGCAGCGTCAGATGTGTATAAGAGACAGCCTACGGGNGGCWGCAG-3′; reverse primer: 5′-GTCTCGTGGGCTCGGAGATGTGTATAAGAGACAGGACTACHVGGGTATCTAATCC-3′) and the 2 × 300 bp paired-end sequencing of the amplicons was performed as described previously [31].

#### 2.7.3. Taxonomy Analyses

Sequences were processed using FROGS (Version 4.0.1 + galaxy 1) [39], a galaxy-supported pipeline, as described previously [31], except that version 138.1 filtered at a pintail score of 80 was used for the SILVA 16S database.

#### 2.7.4. Statistical Analyses of the Diversity and Structure of Caecal Microbiota

The diversity and structure of caecal microbiota were analysed using the phyloseq R package implemented in FROGS, as described previously [31]. The richness, Chao1 richness, and the Shannon and InvSimpson indices were used to describe the diversity in samples (richness and evenness) (Appendix A). An ANOVA was followed by Tukey’s HSD test to investigate the effect of vaccination on these indices. The impact of vaccination on microbiota composition and structure was also observed. A Bray–Curtis distance matrix, which takes into account the relative abundance of OTUs shared by different samples, was calculated after data rarefaction and plotted using multi-dimensional scaling (MDS) to investigate the structure of the bacterial community. The significance was checked by an ADONIS pairwise test. The non-parametric Kruskal–Wallis test was used to compare the relative abundance of the major phyla and nine main genera between groups and the Wilcoxon test was used to estimate significant differences (*p* ≤ 0.05) between groups. The linear discriminant analysis (LDA) effect size (LEfSe) method was applied with an LDA > 2 to identify taxa with a differential abundance (Appendix A) that was statistically different between groups.

## 3. Results

### 3.1. Body Weight and Clinical Observations

Each bird was individually weighed during the trial. There was no effect of vaccination and adjuvants on chicken growth, as no difference in mean body weight was observed between each group tested from D5 to D42 compared with the challenge group (*p* > 0.05) (except for the YP437 I2, P I2 and YP 437 I4 groups on D12 and the P I2 group on D19 and D22) (Table 1).

### 3.2. Campylobacter Caecal and Faecal Enumeration

*Campylobacter* enumerations were assessed from caecal contents on D42, when there was a significantly greater *Campylobacter* colonisation level (*p* ≤ 0.05) in the YP437 I4 and P I4 groups than in the YP437 I2 group (Figure 2). However, an elevated inter-individual variability was observed in all the tested groups, ranging from 2.00 to 8.56 log_10_ CFU/g (Figure 2). It is interesting to note that when performing a T-test comparing only YP437 I2 and P I2, the *p* value was 0.04, which means that in that case, the *Campylobacter* load was lower in the YP437 I2 group than in the P I2 one. Likewise, it appeared that the YP437 I2 group tended to have a lower *Campylobacter* load than the challenge one (tendency was shown (0.05 > *p* > 0.1); *p* = 0.09). These results indicate that, at the very least, there appears to be a tendency for the YP437 I2 group to have a lower *Campylobacter* load.

### 3.3. Specific Serum and Bile Anti-YP Antibodies by Specific ELISAs

The levels of specific systemic antibodies in serum (IgY) and specific mucosal antibodies in bile (IgA) against the YP437 antigen were assessed with ELISAs in all groups on D19 and D42 (Figure 3).

First, the levels of specific anti-YP437 IgY in serum were numerically greater on D19 in the two vaccinated groups inoculated intramuscularly (YP437 I2 and YP437 I4) than in either of their placebo groups (P I2 and P I4) or the challenge group, but the difference was not significant (*p* > 0.05). On D42, the levels of specific anti-YP437 IgY were significantly greater (*p* ≤ 0.05) in both the YP437 I2 and YP437 I4 groups than in their respective placebo groups and the challenge group. More specifically, the level of IgY was significantly greater (*p* ≤ 0.05) in the YP437 I4 group than in the YP437 I2 group, demonstrating that four inoculations of the YP437 vaccine induced greater levels of specific IgY than two inoculations in chickens on D42.

On D19, the production of specific anti-YP437 IgA in bile was greater in the YP437 I4 group than in the other tested groups, but the difference between its respective placebo group (P I4), YP437 I2, and the challenge groups was not significant (*p* > 0.05). On D42, a significantly (*p* ≤ 0.05) greater production of specific anti-YP437 IgA was observed in the YP437 I4 groups than in the other groups. However, an elevated inter-individual variability was observed in all groups, mainly for IgA on D42, with ODs_490nm_ ranging from 0.00 to 1.86 (Figure 3).

### 3.4. Caecal Microbiota Analyses

The effect of vaccination with the YP437 vaccine, inoculated two or four times intramuscularly, on the caecal microbiota was assessed by analysing the caecal content of the chickens in the different groups (YP437 I2, P I2, YP437 I4, P I4 and challenge) on D42 using 16S rRNA metabarcoding. For the challenge group, six chickens were picked randomly for the analysis, whereas in the other groups, all the chickens euthanised were analysed (Figure 1).

The number of sequences obtained from 72 samples after read demultiplexing and pre-processing (merging, denoising and dereplication) was 3,060,714, with a median of 48,946 sequences per sample (minimum: 16,410; maximum: 87,620).

A total number of 922 OTUs were obtained with 1,286,238 sequences; the minimum and maximum number of OTUs in each group are indicated in Table 2.

The alpha and beta diversity of the caecal microbiota were assessed (Figure 4 and Figure 5). Both the richness (observed species) and the diversity (Shannon and inverse Simpson indices) indices were compared between groups for the alpha diversity found within samples (Figure 4). No significant difference (*p* > 0.05) in richness was observed between the groups, whereas the diversity was greater (*p* ≤ 0.05) in the challenge group than in the YP437 I2 and P I2 groups with the Shannon index and greater than in the P I2 group with the inverse Simpson index (Figure 4). These results suggest that the groups contained the same number of species but that these species were more evenly distributed in the challenge group than in the YP437 I2 and P I2 groups.

To study the beta diversity, the differences in microbial population structures among the groups were examined by MDS based on the Bray–Curtis distance (Figure 5a). A clear segregation of the YP437 I2 and YP I2 groups from the YP437 I4, P I4 and challenge groups was observed (Figure 5a). These results suggest that OTUs with the greatest relative abundance in the YP437 I2 and P I2 groups differ from those in the YP437 I4, P I4 and challenge groups. The heatmap, which represents the community structure with the relative abundance of OTUs, also confirmed the segregations between the YP437 I2 and YP I2 groups and the YP437 I4, P I4 and challenge groups (Figure 5b). Multivariate ANOVA (performed with Adonis) revealed significant (*p*  ≤  0.001) differences in the bacterial community structure between groups and the group effect explained 43% of the observed variation.

The caecal bacterial composition in each group at the phylum level on D42 is represented in Figure 6. The *Firmicutes* phylum predominated in each group, followed by the *Proteobacteria*, *Campylobacterota* and *Actinobacteriota* phyla (Figure 6a). The relative abundance of the *Firmicutes* phylum was significantly greater (*p* ≤ 0.05) in the P I2 group than in the P I4 group but not in the YP437 I2, YP437 I4 and challenge groups (*p* > 0.05). The *Campylobacterota* phylum was significantly greater in P I4 than in YP437 I2 (*p* ≤ 0.05) but not in the other tested groups (*p* > 0.05) (Figure 6b). The *Actinobacteriota* phylum was significantly greater in the P I4 and challenge groups than in the YP437 I2 and P I2 groups (*p* ≤ 0.05) but not than in YP437 I4 (*p* > 0.05) (Figure 6b). These results suggest that *Campylobacter* alone without treatment increases the relative abundance of *Actinobacteriota*.

The caecal bacterial composition in each group at the genus level (nine main genera) on D42 is represented in Figure 7. The relative abundance of the *Faecalibacterium* genus was significantly greater (*p* ≤ 0.05) in the YP437 I2 and P I2 groups than in the YP 437 I4, P I4 and challenge groups. On the contrary, the relative abundance of *Blautia* was significantly greater (*p* ≤ 0.05) in the YP 437 I4, P I4 and challenge groups than in the YP437 I2 and P I2 groups. The relative abundance of *Oscillibacter* appeared greater in the YP437 I2 and P I2 groups than in the YP 437 I4, P I4 and challenge groups, whereas the relative abundance of *Subdogralinum* appeared greater in the YP437 I4, P I4 and challenge groups than in the YP 437 and P I2 groups, but these differences were not significant for all the groups. Moreover, the relative abundance of *Colidextribacter* was significantly greater (*p* ≤ 0.05) in the YP 437 I4 group than in the P I2 group but not in the other groups tested. However, an elevated inter-individual variability was observed in all the groups tested (Figure 7b). It should be mentioned that considering these nine genera, no significant difference in relative abundance was observed between the vaccinated groups and their respective placebo, indicating that the microbiota modifications observed are not due to the vaccine protein (Figure 7b). In addition, no significant difference was observed between the YP 437 I4 and P I4 groups and the challenge group (Figure 7b). These results are in accordance with the results observed for beta diversity.

In addition, a linear discriminant analysis (LDA) effect size (LEfSe) algorithm approach identified the taxa characterising the differences between the groups. However, no specific taxa were identified (Appendix A), indicating that no specific OTUs with a statistically different relative abundance between the groups were identified.

## 4. Discussion

The risk of the incidence of human campylobacteriosis could be reduced by vaccinating broilers against *Campylobacter*. Despite numerous studies on such vaccination in the past few decades, this control strategy is still under development. In fact, the results of many vaccine research studies have been inconclusive and/or inconsistent and the mechanisms of protection against *Campylobacter* following vaccination are still not fully understood.

In our previous study [33], the YP437 vaccine candidate was identified by reverse vaccinology as a potential vaccine candidate against *Campylobacter* and, using a DNA prime/protein boost vaccine regimen, this vaccine candidate induced a partially protective immune response against *Campylobacter* [34]. However, the protections were not confirmed in a second trial. Nevertheless, since this vaccine candidate induced partial protection once, it was tested again to determine which vaccine parameters may be improved to develop an effective vaccine against *Campylobacter* in broilers. A protein prime/protein boost vaccine regimen with the YP437 vaccine candidate had not been tested until the present study. In fact, according to our previous study, the inoculation of proteins could be an alternative to the use of a DNA vaccine [40] and, according to another team, two inoculations of only proteins (e.g., FlaA, FlpA and CadF) by the intramuscular pathway has already demonstrated a significant reduction in caecal *Campylobacter* [41].

The YP437 vaccine candidate was thus tested using a protein prime/protein boost regimen, with two or four vaccine injections, to evaluate whether boosters could increase protective immune responses. Two or four intramuscular inoculations of the YP437 protein (YP437 I2 and YP437 I4 groups) induced a greater specific systemic immune response (production of specific IgY anti-YP437) than in the placebos; more specifically, the immune response was even greater with four inoculations (YP437 I4 group) than with two (YP437 I2 group). In bile, a significant specific mucosal immune response (production of specific IgA anti-YP437) was observed only in the YP437 I4 group on D42 compared with the other groups tested. These results demonstrate that intramuscular vaccination with the YP437 vaccine was effective in inducing an antibody response. More specifically, the frequency of inoculation had an impact and, as expected, four inoculations induced a stronger antibody response than two inoculations. However, no clear effect on *Campylobacter* caecal load was observed in these two vaccinated groups, despite a strong systemic and mucosal immune response observed in the YP437 I4 group, as already observed using the DNA prime/protein boost regimen [34]. The absence of protection could be explained, as already suggested, by the fact that IgY might have been absent or inaccessible (degraded, denatured, or both), during the passage through the intestines, thereby reaching the caecum at insufficient concentrations, or might not be active in the caecum [16,42,43]. Moreover, it could be hypothesised that in the YP437 I4 group, the antibodies produced following the vaccinations of YP437 on D5 and D12 could bind the protein inoculated on D19 and D28 rather than binding to the bacteria challenged.

As seen in the previous study using a DNA prime/protein boost vaccine regimen against *Campylobacter* [35], a slight change in the microbiota was observed after vaccination. The results of this study suggest that differences between the groups reflect an effect of the frequency of inoculation or adjuvant rather than vaccination with the YP437 protein. The YP437 I4, the P I4 and the challenge groups appeared to be associated with a greater relative abundance of the *Actinobacteriota* phylum and *Blautia* and *Subdogralinum* genera, whereas the YP437 I2 and P I2 groups appeared to be associated with a greater relative abundance of *Faecalibacterium*. *Blautia*, *Subdoligralinum* and *Faecalibacterium*, which are short-chain fatty acid producers and are part of the core microbiome found in the caeca of healthy chickens [44]. A previous study with a DNA prime/protein boost flagellin-based vaccine revealed an increase in *Faecalibacterium* and *Blautia* in response to vaccination against *Campylobacter* [31].

Although the vaccination of broiler chicks is an attractive approach for controlling *Campylobacter* colonisation, there remain immunological and logistical barriers that must be overcome. Indeed, the immune function is limited in the first 2–3 weeks post-hatch because maternal antibodies could block interactions between circulating pathogen-derived antigens and immune cells [45]. Moreover, an immature bursa of Fabricius leads to an immature antibody repertoire until 5–7 weeks [46]. Consequently, there is a limited time window for the induction of immune responses, since broilers are slaughtered at 5–6 weeks of age. Moreover, the role of antibodies in the reduction of *Campylobacter* is not clear. Indeed, elevated antibody levels (IgY or IgA) do not necessarily result in effective protection among broilers, as already observed previously after a challenge by *Campylobacter* only [47] or in the case of a *Campylobacter* challenge with the use of different proteins as a vaccine [31,34,35,48,49]. On the contrary, humoral immunity, more specifically the intestinal mucosal immune response, appears to be involved in the reduction of *Campylobacter* in chickens [12,21,50].

The vaccination regimen used in this study could be refined by testing another approach, for example, another pathway such as the in ovo route. In ovo vaccination has already demonstrated a protective role [13,16,17,51]. However, trials failed to trigger a significant mucosal immune response and broilers were not protected against *C. jejuni* infection. Despite showing only a moderate reduction in *Campylobacter* load, these results suggest that in ovo immunisation could be a beneficial strategy that could potentially be enhanced with booster vaccinations post-hatching.

Oral vaccination could be also tested as it could induce a local immune response on the site of *Campylobacter* colonisation. Moreover, it would reduce the cost of vaccine administration as well as the safety risks and stress among animals. Promising results were recently observed against *Campylobacter* [14,22,52,53,54,55]. Thus, the use of antigens delivered by microorganisms or a nanoparticle strategy could be advantageous for subsequent trials. However, mucosal vaccines could face several challenges such as dilution and entrapment by mucosal secretions, degradation by proteases, and exclusion by the epithelial barriers [56]. As reviewed in [56], future studies need to advance formulation development and vaccine delivery technology as well as investigate immunological mechanisms of oral vaccine delivery systems.

The difficulties we encountered in this trial are related to a vaccine antigen that did not induce immune responses that were effective in reducing *Campylobacter* in the caecum. Consequently, it appears necessary to identify new antigen candidates or to combine several antigens or epitopes. In addition to the YP437 antigen used in this study, other vaccine candidates have been identified by this method and could be used in forthcoming trials [33]. The strategy of reverse vaccinology performed in 2014 and published in 2016 used only the *Campylobacter jejuni* subsp. *jejuni* 81-176 genome available in the Vaxign database to identify antigens which could be potential vaccine candidates against *Campylobacter* [33]. However, the sequence of a single genome does not reflect genetic variabilities within a bacterial species and limits genome-wide screens for new vaccine candidates [57]. Recently, new bioinformatics tools have been developed and new methods have appeared. The pangenomic reverse vaccinology method, based on a genome-wide reverse vaccinology model using the global genetic repertoire of species, increases the possibility of identifying new vaccine candidates [57]. This strategy has already been used in chickens against other bacteria such as *Gallibacterium anatis* with the use of ten genomes [58] and avian pathogenic *Escherichia coli* with the use of 58 and 127 genomes in two studies by the same team [59,60]. Because of the elevated genetic diversity across *Campylobacter* serotypes [61], a genome-wide reverse vaccinology model using the global genetic repertoire of thermotolerant *Campylobacter* species responsible for campylobacteriosis could be applied. Indeed, a recent study focused on the genome sequences of four *C. jejuni* strains from different sources. The study’s in silico genome analysis predicted three conserved potential vaccine candidates: phospholipase A (PldA), TonB-dependent vitamin B12 transporter (BtuB), and cytolethal distending toxin subunit B (CdtB) [62]. This method could be applied in the future to a larger number of *Campylobacter* genomes.

Moreover, a multi-epitope vaccine could be developed thanks to reverse vaccinology and immunoinformatics tools. This method has previously been used against the avian pathogen *Mycoplasma gallisepticum*, which causes chronic respiratory disease [63]. Two studies have reported the development of a multi-epitope vaccine against *Campylobacter* [64,65].

## 5. Conclusions

This study reveals that, although the intramuscularly inoculated protein-based vaccines induced humoral and mucosal immune responses, no significant reduction in *Campylobacter* caecal load was observed. This means that immune responses alone are not sufficient to offer broilers protection against *Campylobacter*.

Further studies are needed to identify the key parameters involved in protection against *Campylobacter*. Moreover, other vaccine candidates, other vaccine vectors and/or other vaccine regimens (e.g., in ovo and mucosal) could be tested in broilers.

## Figures and Tables

**Figure 1 animals-13-03779-f001:**
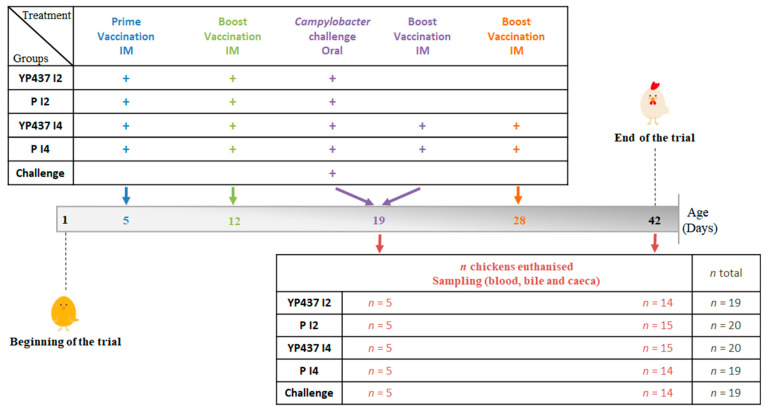
Summary of the experimental procedure. The number of chickens (*n*) necropsied per group on each euthanisation day is shown. For each chicken, 100 µg of recombinant YP437 protein for the vaccinated groups or phosphate-buffered saline (PBS) for the placebo groups was emulsified in MONTANIDETM ISA 78 VG (37/63, *w*/*w*). IM: intramuscular route.

**Figure 2 animals-13-03779-f002:**
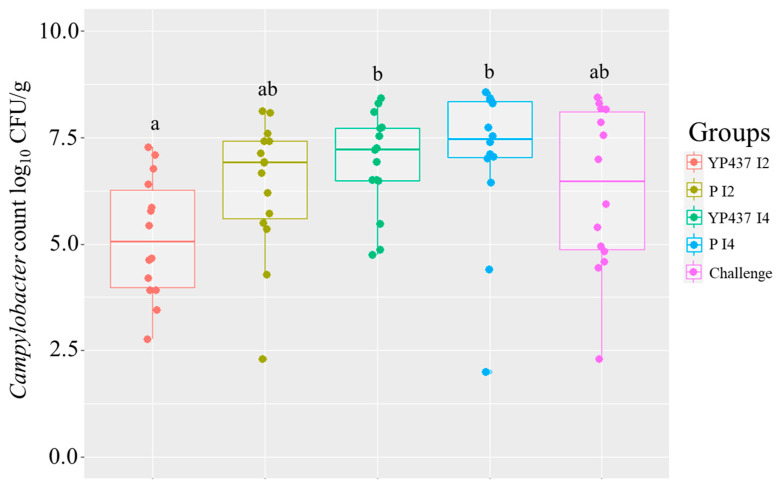
Caecal *Campylobacter* load on D42. Each point represents one chicken. The inter-group caecal *Campylobacter* load was compared using the non-parametric Kruskal–Wallis test. Superscript letters represent significant differences (*p* ≤ 0.05) between groups estimated using the Wilcoxon test.

**Figure 3 animals-13-03779-f003:**
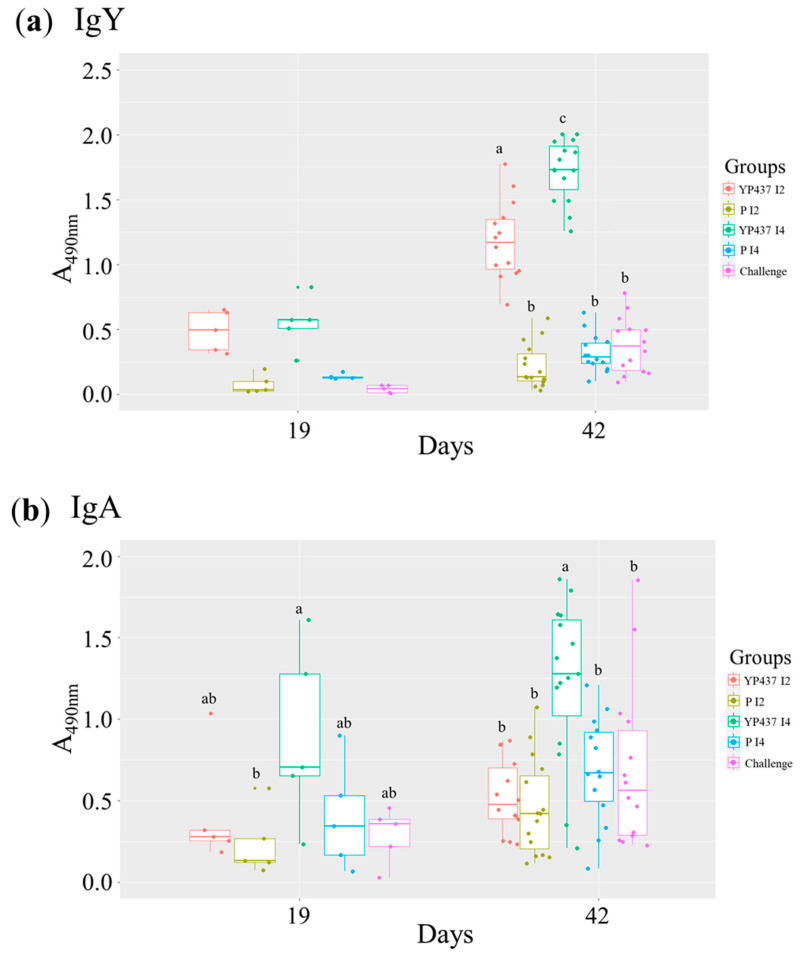
Specific anti-YP437 levels (IgY in serum and IgA in bile) on D19 and D42 in all groups. Each point represents one chicken. (**a**) Specific anti-YP437 IgY in serum. (**b**) Specific anti-YP437 IgA in bile. A parametric ANOVA test was used to compare antibody levels between the groups on D19 and D42 when the normality and homogeneity criteria of the variances were validated; otherwise, the non-parametric Kruskal–Wallis test was applied. Superscript letters represent significant differences (*p* ≤ 0.05) between groups (estimated using the Tukey test after an ANOVA parametric test or the Wilcoxon test after the Kruskal–Wallis test).

**Figure 4 animals-13-03779-f004:**
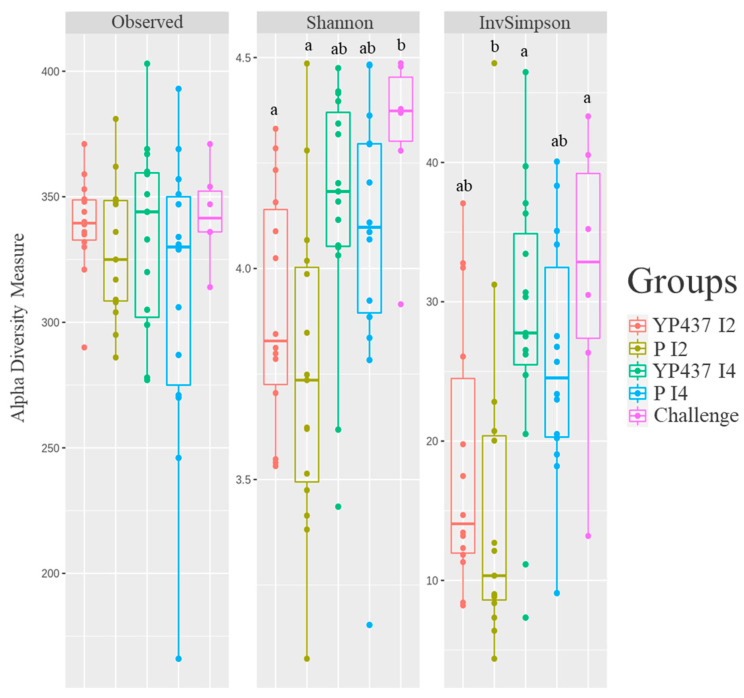
Richness and alpha diversity indices for the chickens’ caecal microbiota on D42. Each point represents one chicken. The parametric ANOVA test was applied to compare the groups. The Tukey test was applied to estimate significant differences (*p* ≤ 0.05) between groups (represented by superscript letters).

**Figure 5 animals-13-03779-f005:**
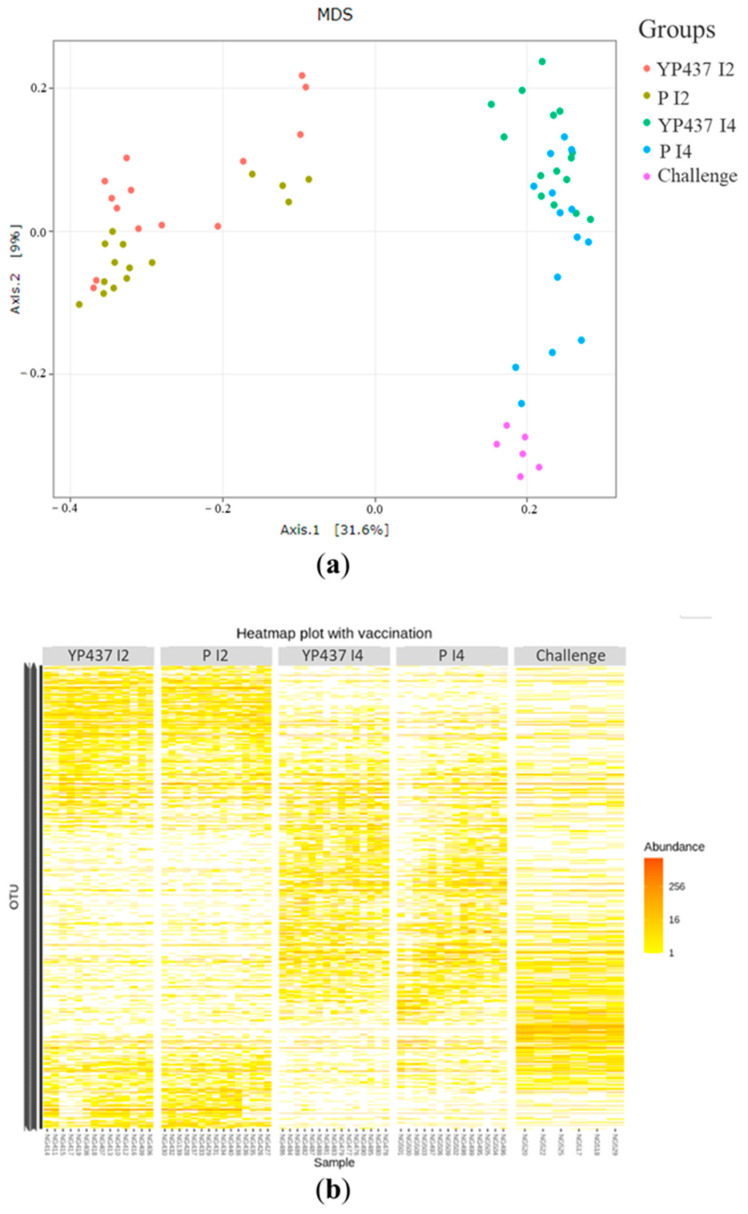
Representation of the beta diversity for the chickens’ caecal microbiota on D42. (**a**) MDS based on Bray–Curtis distance. Each point represents one chicken. (**b**) Heatmap representing the community structure. The relative abundance of OTUs is represented by the colour scale, with yellow being the least and red the most abundant.

**Figure 6 animals-13-03779-f006:**
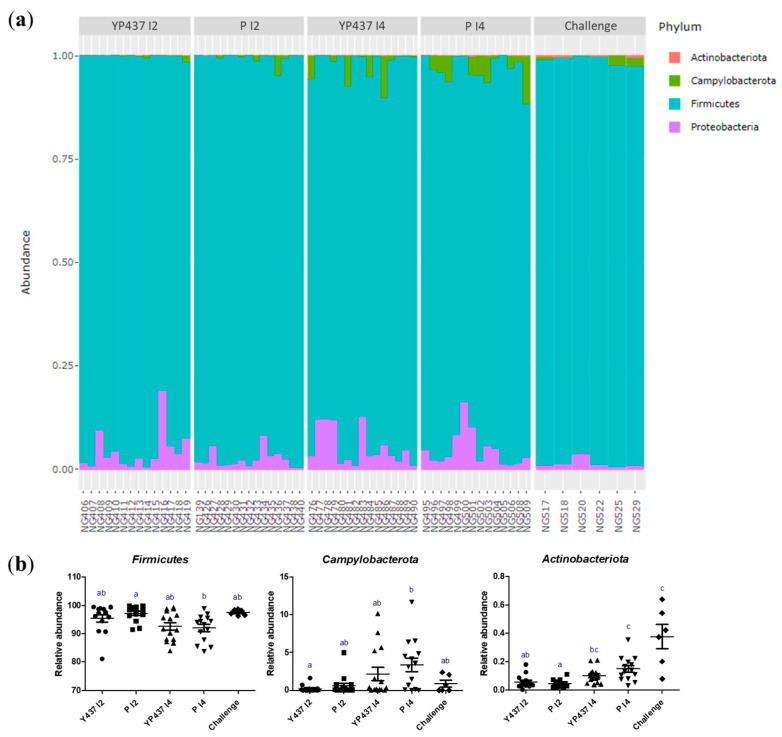
Composition of the caecal microbiota at the phylum level on D42. (**a**) Relative abundance of the four phyla identified in caecal microbiota represented by a bar for each sample. (**b**) Relative abundance of the three phyla presenting significant differences between groups. Each point represents one chicken. The non-parametric Kruskal–Wallis test was used to compare the groups. Superscript letters represent significant differences (*p* ≤ 0.05) between groups (estimated using the Wilcoxon test).

**Figure 7 animals-13-03779-f007:**
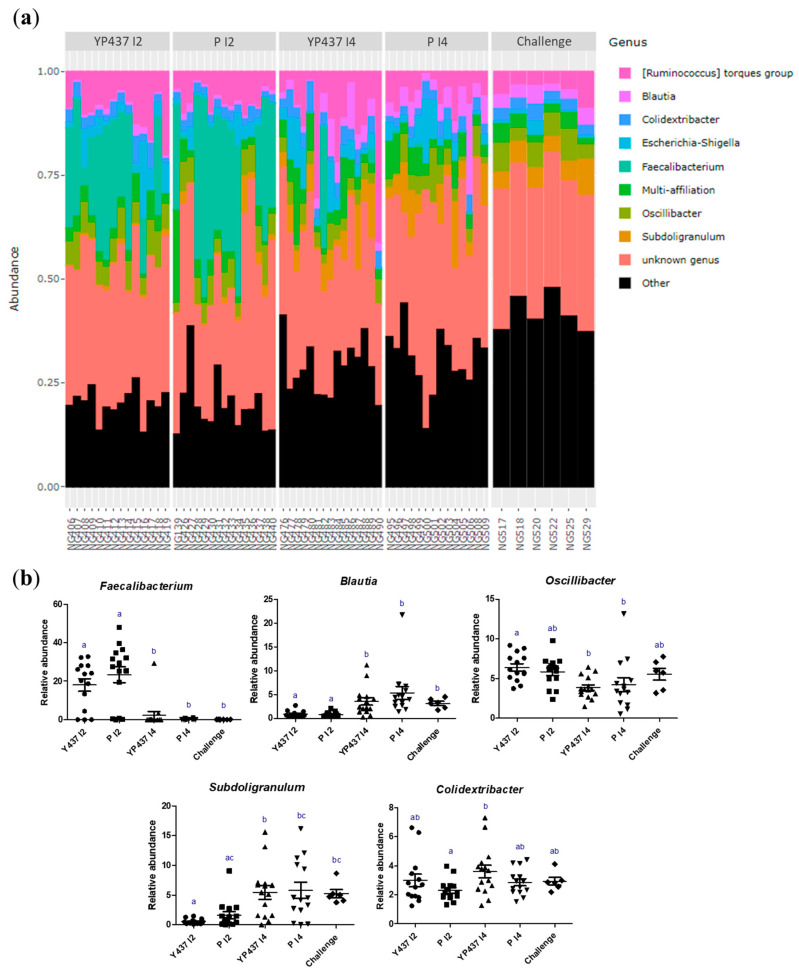
Composition of the caecal microbiota at the genus level. (**a**) Relative abundance of the nine main genera. Each bar represents one sample. (**b**) Relative abundance of the major genera with a significant difference between the groups. Each point represents one chicken. The non-parametric Kruskal–Wallis test was applied to compare the groups. Superscript letters represent significant differences (*p* ≤ 0.05) between groups (estimated using the Wilcoxon test).

**Table 1 animals-13-03779-t001:** Body weights (mean ± SD in g) of chicken groups during the trial. An ANOVA parametric test was used to compare body weights between the different groups each day when the normality and homogeneity criteria of the variances were validated; otherwise, the non-parametric Kruskal–Wallis test was applied. Significant differences (*p* ≤ 0.05) between each group were estimated using the Tukey test after an ANOVA parametric test or the Wilcoxon test after the Kruskal–Wallis test. A significant difference between the challenge group and another group is indicated by *.

Groups	Day 5	Day 12	Day 19	Day 22	Day 28	Day 42
YP437 I2	113 ± 8	365 ± 27 *	775 ± 52	1041 ± 85	1656 ± 148	3381 ± 373
P I2	111 ± 12	363 ± 34 *	754 ± 64 *	966 ± 107 *	1608 ± 191	3164 ± 545
YP437 I4	113 ± 10	369 ± 30 *	775 ± 71	1002 ± 110	1557 ± 214	3023 ± 424
P I4	111 ± 8	383 ± 29	802 ± 66	1061 ± 95	1668 ± 178	3292 ± 483
Challenge	112 ± 6	396 ± 23	822 ± 70	1084 ± 85	1740 ± 156	3416 ± 418

**Table 2 animals-13-03779-t002:** Minimum and maximum OTUs per group.

Groups	OTUs: Max	OTUs: Min
YP437 I2	531	409
P I2	554	403
YP437 I4	597	388
P I4	563	219
Challenge	542	420

## Data Availability

The data presented in this study are available on request from the corresponding author.

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
