# Peer review of "Evaluation of Two Recombinant Protein-Based Vaccine Regimens against Campylobacter jejuni: Impact on Protection, Humoral Immune Responses and Gut Microbiota in Broilers"

_animals, 2023, doi:10.3390/ani13243779_

Round 1
Reviewer 1 Report (New Reviewer)
Comments and Suggestions for Authors
The authors describe potential vaccine candidates against Camphlobacter jejuni in broiler chickens. Although, the vaccines didn't reduce Camphlobacter caecal load, the manuscript is usual toward development of future vaccines.
The authors analyzed the immune response following vaccination using an ELISA test. Were the antisera tested for neutralization of the native Camphlobacter species used for challenge? The authors did mention in the discussion on lines 426-428, that the antibodies produced following vaccination bound the protein used for inoculations, however, may not bind the challenge bacteria. It would be important to do some bioassays to evaluate the ability of the antibodies produced following vaccination with the proteins for ability to bind and neutralize the bacterial species.
The authors did mention the evaluation of in ovo vaccinations. Since a majority of the vaccination regimen in the broiler industry relies upon in ovo vaccination with subsequent boosters, this would be an important step towards evaluating the validity of their protein vaccine candidate.
Author Response
We would like to thank the reviewer for his valuable comments
Please see the attachment.

Reviewer 2 Report (New Reviewer)
Comments and Suggestions for Authors
The manuscript needs only some changes to improve it.
The day D19 is a critical time. Make it clear in Fig. 1, that both the C. challenge and the boost vaccination were carried out in D19 (two arrows or brace). In lines 136-145 please clarify the time schedule of D19: How were the challenge, vaccination, sampling organized to avoid cross-contamination?
lines 150-154: How was the outer Campylobacter contamination excluded from the experimental groups by the litter, commercial feed, water etc.?
lines 443-481: Comparison and analysis of alternative vaccination methods are far more than it is required as the discussion of the current results. It should be reduced, strict to the results. Nevertheless, these remarks are correct, can be part of a review article. Instead; the role and importance of 'the nine main genera' (Fig 7a) should be detailed in Discussion extending the very brief remarks in line 434-441.
Comments on the Quality of English Language
The English language usage is correct.
Author Response
We would like to thank the reviewer for his valuable comments
Please see the attachment.

Reviewer 3 Report (New Reviewer)
Comments and Suggestions for Authors
The aim of this study was to evaluate of two recombinant protein based vaccine regimens against Campylobacter jejuni on humaral immune response and gut microbiota in broilers. The obtained research results are important for poultry producers and consumers of chicken meat. The main reservoir of Campylobacter is poultry. Campylobacter jejuni is the most common cause of bacterial diarrhea in humans in industrialized countries. Campylobacteriosis is especially dangerous for people with chronic diseases.
General concept comments:
In my opinion, the article should be supplemented with the following information:
- what immunoprophylaxis program was implemented, were there vaccinations against IB, IBD, MD?
- what was the type of building in which the research was carried out - closed, without windows?, what was the color and intensity of light?
- What was the feeding program?
- What was the feed intake (FI) - body weight depends on IF; feed conversion ratio? and above all, mortality in the studied periods?
- Were the concentrations of C02 and NH3 monitored and, if so, did they exceed the standards?
Specific comments
L132 (2017) what number?
L135 what type of building?
L136 Were the experimental vaccines against Campylobacter the only ones used during broiler chicken rearing?
L136 What about the use of vaccines for IB, IBD, MD?
L136 Why 0.3mL dose?
L144 (Total = …..) how many birds were weighed in total
L149 What concentration of C02 and NH3 was measured?
Table 1 Please use the letters "a, b" to indicate significant differences - as in Figures
Figure 3a and 3b please increase the size
L371 P4 or P14?
L520 delete "6. "Patents"
L562 "Br. Poult. Sci.” instead of current form according to ISO 4
L716 Poult. Sci. instead of current form
Comments on the preparation of the article
Please prepare the article in accordance with the instructions for authors:
• For significance please use lowercase "p" in italic instead of uppercase "P" throughout the main article
• In References chapter please use a "dot" after each abbreviation, for example J. Appl. Microbial. instead of J Appl Microbial
Author Response
We would like to thank the reviewer for his valuable comments
Please see the attachment.

This manuscript is a resubmission of an earlier submission. The following is a list of the peer review reports and author responses from that submission.
Round 1
Reviewer 1 Report
Comments and Suggestions for Authors
The manuscript submitted for review describes novel recombinant, protein-based vaccination regimes for the reduction of Campylobacter colonization in poultry. Although the study is generally well-written and combines several appropriate methods to test these vaccine regimes, the overall outcome has minimal utility in the advancement of campylobacter vaccine research. My overall impression was that administering 4 intramuscular vaccines to broilers in a small 3-5 week window before slaughter will not be practical in a commercial setting, assuming the vaccine is efficacious. None of the described regimes were efficacious in reducing campylobacter loads. These findings diminish potential impacts of the study despite observed differences in microbiota composition and host immune response. I am recommending rejection so this can consolidated and submitted as a research note or brief report instead of a full length article. Should the authors seek publication as a full length article, they should consider testing this vaccine in ovo as described in the discussion as this approach is much more practical from a production perspective.
Should you choose to resubmit, I have attached a list of comments and suggested edits to the manuscript.

Comments on the Quality of English Language
There were moderate corrections to the syntax, which are indicated in the attached revision document.
Reviewer 2 Report
Comments and Suggestions for Authors
Line 63, check the grammar and spelling.
Line 134, In which muscle was the injection done?
Three stages of vaccine injection are not possible in practical conditions, and it is also problematic in terms of ethics and animal welfare.
Line 174, The number of replications and observations, is not clear.
Were the birds raised on the litter or in the cage? Because access to feces will affect the trial results.
Table 1, Contrary to expectations, the challenged group has a higher weight in 42 days, why?
Table 1, There is a need to provide one of the deviation criteria. Because the differences are considerable, but not statistically significant.
It should be explained to the reader of the article whether the contamination is transmitted to humans through the blood or edible tissues of the bird, or whether the contamination is transmitted from the cecum to the carcass in the slaughterhouse. If it is the second, then what is the role of the vaccine and humoral immunity in this matter.
Comments on the Quality of English Language
Line 63, check the grammar and spelling.
There are extra spaces between some words.
Reviewer 3 Report
Comments and Suggestions for Authors
It is an interesting manuscript. However, it has some issues. 1) Please check microbiota taxa abundance. Phylum level: Where is Bacteroidetes? Family level: What is Multiaffiliation group?; 2) Please check the statistics; 3) also describe the sampling and replicates
Line 3: immune responses, and gut microbiota
Line 19: I suggest: but vaccines for Campylobacter is not available to date.
Line 20: Is it your previous work? Please indicate “in our previous” or if it was others, please remove this. There should be no reference in the abstract.
Line 28: Remove “for example by..”
Line 31: Vaccines for campylobacter?
Line 31: In a previous work? Is it yours or other? if it was others, please remove this. There should be no reference in the abstract.
Line 33: Please rewrite this: In order to research ways of improving vaccine efficacy. I don’t understand what this is.
Line 33: I think this sentence is out of focus. Is this sentence related to : In order to research ways of improving vaccine efficacy, the vaccination protocol was modified using a protein prime/protein boost regimen with a different number of boosters.
Line 35: Broilers were given two or four intramuscular protein vaccinations (e.g., YP437 vaccine antigen) before an oral challenge 36 by C. jejuni during a 42-day trial.
Line 37: By T-test? Which stastistical method did you use?
Line 40: but no reduction in Campylobacter caecal load was observed in any 40 of the groups. Please value (P > 0.05)?
Line 41: Abstract should be different from the simple summary. Please be specific. “Which parameter was reduced compared to what” + P value. So was it from the beta diversity analysis?
Line 44: I do not get it about the conclusion. Why do you need to go back to find other vaccines? I thought you did this study because your vaccine candidate was effective in your previous work?
You should provide the conclusion based on what you observed. For example, the candidate vaccines were not effective to induce humoral immune response, therefore, the candidate vaccines did not provide protection against campylobacter infection in broiler chickens. More studies are required to find new candidate?
Line 48: I think it should be campylobacter spp. or campylobacter jejuni (be specific)
Line 56: You mentioned only broiler. Maybe it would be better to say broiler meat products in the entire manuscript? Poultry is too broad: broiler, turkey, layers..
Line 59: Please clarify 8 log value is entire bacteria or campylobacter. Please be specific.
Line 61: chicken reservoir, and more specifically in the caecum,-> in the ceca of chickens. Remove “therefore”
Line 60 to 63:
I suggest : reducing Campylobacter in the ceca of broiler chickens would be a effective way to reduce the risk to consumers and the public health burden, which has been 62 estimated at about 2.4 billion euros each year in the European Union based on 2011 figures 63 [7].
Line 64: This part does not make sense for me. “the relative European Union risk of human campylobacteriosis attributable to broiler meat by 42% and 58%, respectively” Could you rewrite it?
Line 66 to 68: These sentences should be placed before line 59. It should be like: 1) It should be effective to control Campy at the farm -> 2) targeting ceca should be effective. Please rewrite as suggested. I believe you can reorganize the sentences better.
Line 72: remove of poultry. Several strategies ? You said vaccination is the most promising. Why do you mention this one after you said vaccination is good strategy? 12 references? For this sentence ? Please reduce the references
Line 73: Initial study..? Our previous study? And please provide the reference
Line 74: reference? And was it campy vaccine? Or ? Was this your study? Otherwise please provide references
Line 76: “According to ~~” . I think this sentence is out of the context. Please remove.
Line 78: Remove “despite all these efforts”
Line 79: remove “sometimes”
Line 80: several studies
Line 82: Six references are too many.
Line 86: research programme? In our previous study?
Name YP437
Line 88: in broiler chickens. ?
Line 92: confirmed -> observed. IF you used confirmed, it indicates you may be biased.
Line 93: Gloanec et al., manuscript currently submitted for publication -> Our unpublished works
Line 95: In which animal? It is very important to indicate.
Objective: Please indicate which animal.
Please change as follows.
Therefore, the objectives of this project were (1) to evaluate the efficacy of the vaccine candidate using a protein prime/protein boost regimen; (2) to assess the impact of the frequency (booster inoculations) of vaccine inoculations; and 3) to study in greater depth the systemic and local humoral immune responses generated and gut microbiota modifications after vaccination.
Materials and methods
Line 109: a few days -> Please clarify
Line 111: What is this ? MONTANIDETM ISA 78 111 VG (37/63, w/w) Please describe what is this. Is it solution? Also this one: an Ultra Turrax® Tube drive Basic (IKA®-Werke GmbH, Staufen, 112 Germany) according
Line 117: Ploufragan? Is it city? Please provide country.
Line 118: Please provide time 24h ? or overnight.
Line 125: an approved facility
Line 126: You meant the animal use was approved or facility was approved? Please indicate it.
Line 130: in randomly selected five chicks according to the standard protocol ?
Line 131: Is it acceptable? 19 to 20 chiks..?
Line 132: into five groups (YP437 I2, P I2, YP437 I4, P I4, and challenge) and were kept in 3.42 m² floor pens (1.85 × 1.85 m2) as described in Figure.
Line 134: day (D)5 and D12. You need a space between the letter and number (e.g., D 12).
Line 136: How much ? 0.5 mL?
Lien 137: 5 to 15 birds. You should be consistent? At least provide exact number for each group. The stocking density definitely can affect the spread of campy infection in a flock.
Line 138: Please describe the method for counting CFU. It is very important.
Line 147: Please provide the CP and energy level. The CP level definitely can affect the colonization of campy.
Line 162: were performed.
Line 185: Please provide more details rather than “as described previously”. Where did you store the samples?
Line 196: Isn’t it more appropriate say “Taxonomy analysis?” Sequence analyses include alpha, beta, etc.
Line 205: ANOVA followed by Tukey’s HSD test was …
Line 206: You meant did you do individual contrast? Or based on TUKEY?
Line 207: Remove “A”. Just “Bray-curtis
Line 223: Based on what? Please clarify based on the results and P values.
Line 229: Tukey test after an ANOVA parametric test or the Wilcoxon test 229 after the Kruskall-Wallis test and are indicated by *. What is this? Did you do TUKEY or WILcoxon test, Kruskall? You have to choose one. There is no “or” in the statistics. Also if you did TUKEY, you have to show a,b,c. Wrong statical method can be the main factor for rejection.
Line 232: Remove D 22 or Three days post inoculation. Please indicate either one.
Line 235: Why didn’t you show the data? Please report it.
Line 236: What do you mean by no different but there was a difference? Please clarify.
Line 239: what is a high inter-individual variability?
Line 244: P = 0.09. If you would like to trends, you have to mentioned “tended to be lower” and in the statistical method, you have to say “Tendency was shown (0.05 < P < 0.1).”
Line 244: These results indicate that at the very least there appears to be a tendency for the YP437 I2 group to have a lower Campylobacter load. What is this? I do not get where you get this?
Line 254: serum IgY and bile IgA. Remove ()
Line 267: the differences
Line 268: You showed 0.09 as tendency but you said P > 0.05. May you should say P > 0.1.
Also P value should be italicized.
Line 273: Is it conclusion? Please remove this.
Line 283: How come there were so many statistical method ? Significant differences (P ≤ 0.05) between groups estimated using the Tukey test after an ANOVA parametric test or the Wilcoxon test after the Kruskall-Wallis test are represented by superscript letters.
Line 289: What is 16s metabarcoding?
Figure 5. Green and blue are hard to distinguish. Could you please change the color or shape ?
Line 337: The phylum Campylobacterorta is right rather than “Campylobacterorta phylum.
Figure 6: You should be consistent with the letter: a is the highest value ? See the firmicutes. I think the bigger values need A.
I do not believe there was no Bacteroidetes. The results you have are too different from the references. This could be the major reason for rejection.
Why there are less replicates for challenge group?
Figure 7: the resolution of figures is too low.
What is Multiaffiliation group?
Line 392: In a previous study -> Is it your study?
Line 395: Please provide the reference ?
Line 400: according to our previous.
Line 417: than two : What is this?
Line 457: recCjAD -> Please check if it was what you meant.
References
Line 625: Infection and immunity? Please make sure the abbreviation
I think the abbreviation should J. Immunol. Res.
Comments on the Quality of English Language
Need to be improved